# NAB 2-Expressing Cancer-Associated Fibroblast Promotes HNSCC Progression

**DOI:** 10.3390/cancers11030388

**Published:** 2019-03-19

**Authors:** So-Young Choi, Su Young Oh, Soo Hyun Kang, Sung-Min Kang, Jinkyung Kim, Heon-Jin Lee, Tae-Geon Kwon, Jin-Wook Kim, Su-Hyung Hong

**Affiliations:** 1Department of Oral and Maxillofacial Surgery, School of Dentistry, Kyungpook National University, Daegu 700-412, Korea; dentalchoi@knu.ac.kr (S.-Y.C.); kwondk@knu.ac.kr (T.-G.K.); vocaleo@knu.ac.kr (J.-W.K.); 2Department of Microbiology and Immunology, School of Dentistry, Kyungpook National University, Daegu 700-412, Korea; oohsuy@knu.ac.kr (S.Y.O.); black_bean@knu.ac.kr (S.H.K.); dkdkdk43@knu.ac.kr (S.-M.K.); kimjk0925@knu.ac.kr (J.K.); heonlee@knu.ac.kr (H.-J.L.)

**Keywords:** head and neck squamous cell carcinoma, NGFI-A-binding protein 2, cancer-associated fibroblast

## Abstract

Cancer-associated fibroblast (CAF)-specific proteins serve as both prognostic biomarkers and targets for anticancer drugs. In this study, we investigated the role of NGFI-A-binding protein (NAB)2 derived from CAFs in the progression of head and neck squamous cell carcinoma (HNSCC). Patient-derived HNSCC and paired metastatic lymph node tissues were examined for NAB2 expression by immunohistochemistry. Primary CAF cultures were established from HNSCC patient tissue, with paired non-tumor fibroblasts (NTFs) serving as a control. CAF or NTF was used to evaluate the effect of NAB2 on HNSCC progression using FaDu cell spheroids and an in vivo mouse xenograft model. NAB2 was detected in interstitial CAFs in primary and metastatic lymph node tissues of HNSCC patients. NAB2 mRNA and protein levels were higher in CAFs as compared to paired NTFs. Conditioned medium (CM) of NAB2-overexpressing CAFs increased the invasion of FaDu spheroids in the Matrigel invasion assay as compared to CM of NTF. Co-injection of NAB2-overexpressing CAFs with FaDu spheroids into mice enhanced the growth of tumors. These data suggest that NAB2-overexpressing CAFs promotes HNSCC progression and is a potential therapeutic target for preventing HNSCC metastasis.

## 1. Introduction

Cancer-associated fibroblasts (CAFs) constitute a sub-population of cells in the cancer stroma with a myofibroblast-like phenotype [1,2] that contribute to tumor progression and are, therefore, prognostic biomarkers [3]. CAFs secrete growth factors and chemokines that alter the extracellular matrix (ECM) and oncogenic signals, thereby increasing the proliferation and invasion of cancer cells [4]; they have also been shown to promote cancer progression in vivo when co-injected with tumor cells or are recruited to the tumor site [5,6]. In addition, irradiated fibroblasts or the medium in which they are grown can enhance the tumorigenicity of breast cancer grafts, which may be a transient effect of CAF-secreted factors [7]. ECM modulators such as matrix metalloproteinases (MMPs) are also regulated by the tumor stroma.

There is increasing evidence that CAFs play an important role in head and neck squamous cell carcinoma (HNSCC) progression [8]. In the tumor stroma, CAFs undergo changes in protein expression including the upregulation of markers such as α-smooth muscle actin (α-SMA) [9,10]. CAFs expressing α-SMA have been detected in several solid tumors including HNSCC and their presence is strongly associated with poor clinical outcome [11]. α-SMA-positive CAFs infiltrate into bone ahead of HNSCC cells by upregulating the receptor activator of nuclear factor kappa-Β ligand in HNSCC [12]. Thus, cancer stromal cells are considered as functional or regulatory cells in the oral cancer microenvironment. Interestingly, CAF-specific expression of the ECM protein periostin and the cell surface glycoprotein cluster of differentiation 44 were also required for the maintenance of cancer stem cell populations in tumors [13,14]. However, it is not well known how CAFs regulate HNSCC progression.

NGFI-A-binding protein (NAB)1 and NAB2 repress transcriptional activation mediated by early growth response 1 (EGR1) [15,16]. While NAB1 is constitutively expressed in most tissues and functions as a general transcriptional regulator, NAB2 is an inducible transcriptional modulator [17]. The human *NAB2* gene is located on chromosome 12ql3.3–14.1, a region that is rearranged in several tumor types [16]. We have observed that NAB2 is specifically expressed in CAFs of HNSCC patient tissue; in this study, we investigated the role of NAB2 on HNSCC progression in vitro and in vivo.

## 2. Results

### 2.1. NAB2 Is Expressed in Cancer-Associated Fibroblasts (CAFs) of Head and Neck Squamous Cell Carcinoma (HNSCC) Patient Tumor Tissues

Samples of primary HNSCC in oral mucosa and metastatic carcinoma in the lymph node were examined for NAB2 expression by immunohistochemistry. There was no immunoreactivity in normal oral mucosa (Figure 1A) or normal lymph node (Figure 1E). In contrast, NAB2 was highly expressed in interstitial CAFs of primary tumor (Figure 1B) and metastatic (Figure 1F) lymph node (red arrows) as compared to their cancer cells (black arrows). NAB2-positive tumor cells and CAFs were counted in primary tumor tissues and metastatic lymph nodes, respectively. As shown in Figure 1D,H, NAB2-positive immunoreactivity was significantly higher in CAFs as compared to tumor cells both in primary tumors and metastatic lymph node tissues. Figure 1C,G are two-fold magnifications of the black rectangle images in Figure 1B and 1F, respectively. Non-tumor fibroblasts (NTFs) were negative for NAB2 expression in both tissue groups.

### 2.2. CAF Marker and Matrix Metalloproteinase (MMP) Expression Is Upregulated in CAFs of HNSCC Patients

We compared CAF marker expression between HNSCC patient fibroblasts that were adjacent to cancer cells (CAFs) and those located in a non-tumor area (non-tumor fibroblasts, NTFs). The mRNA and protein levels of CAF-specific markers such as α-SMA (α–smooth muscle actin) and FAP (fibroblast activation protein) were higher in CAFs than in NTFs from all three patients (P1, P2, P3) (Figure 2A). Similar trends were observed for MMP2, MMP9, and MMP14, which play an important role in HNSCC progression (Figure 2B).

### 2.3. NAB2 and CAF Marker Expression Levels in Fibroblasts Are Correlated

NAB2 mRNA and protein were more highly expressed in CAFs than in NTFs of HNSCC patients (Figure 3A). We compared NAB2 mRNA expression in primary CAF and paired HNSCC tissues. As shown in Figure 3B, NAB2 mRNA level is higher in CAF, supporting the immunohistochemistry (IHC)data from HNSCC tissues (Figure 1). Based on this observation, we evaluated the effect of transient NAB2 overexpression on CAF marker and MMP levels in P3 patients’ fibroblasts (Figure 3C). NAB2 overexpression in NTFs increased CAF marker expression at the mRNA (Figure 3D) and protein level (Figure 3E). A similar effect was observed for MMP (Figure 3F). Conversely, siNAB2 transfection in CAFs reduced CAF marker and MMP mRNA and protein expression (Figure 3G).

### 2.4. NAB2 Expressed by CAFs Enhances FaDu Cell Invasion

We evaluated the effect of NAB2 expressed by CAFs from P3 on FaDu cell invasion using a transwell co-culture system (Figure 4A). The invasion of FaDu cells was markedly increased under co-culture with NAB2-overexpressing CAFs as compared to control vector-transfected CAFs (Figure 4B). On the other hand, FaDu cell invasion was inhibited by co-culturing with CAFs transfected with siNAB2 as compared to control siRNA.

We next investigated the effect of conditioned medium (CM) from CAFs transfected with NAB2 vector or siNAB2 on the invasive potential of FaDu spheroids (Figure 4C). There was no change in spheroid invasion for 3 d; however, on day 14, invasion was enhanced by CM from NAB2-overexpressing as compared to vector control CAFs. The opposite was true for spheroids exposed to siNAB2- as compared to control siRNA-transfected CAFs. Corresponding differences were observed in the number and length of sprouts extending from spheroids (Figure 4D). MMP mRNA expression was up- and down-regulated in FaDu spheroids upon NAB2 overexpression and knockdown, respectively (Figure 4E).

### 2.5. NAB2 Expressed by CAFs Increases the Tumorigenicity of FaDu Spheroids In Vivo

To evaluate the effect of CAF-derived NAB2 on FaDu cell progression in vivo, we transplanted FaDu spheroids along with P3 CAFs transfected with NAB2 or a control vector into nude mice and evaluated tumor growth over time. The growth of FaDu spheroid-derived tumors was enhanced by co-injection of NAB2-overexpressing CAFs (right cheek) as compared to control vector-transfected CAFs (left cheek) (Figure 5A). Hematoxylin and eosin (H&E) staining revealed that tumors arising from spheroids co-injected with NAB2-overexpressing CAFs (Figure 5B, 1R and 5R) were more lobulated (red arrows) and showed increased invasion into the surrounding connective tissue (black arrows) than those derived from spheroids co-injected with control-transfected CAFs (Figure 5B, 1L and 5L). Additionally, while the tumor boundary in the former group was ambiguous, in the latter mice tumors were surrounded clearly by connective tissue. Accordingly, the mRNA expression of genes associated with cancer invasion was increased in tumors in the right cheek (Figure 5C), suggesting that NAB2 overexpressed by CAFs enhances FaDu cell progression in vivo. Figure 6 shows the putative schematic of HNSCC progression by NAB2 derived from CAF.

## 3. Discussion

In recent years, it has been known that the targets of several drugs are stromal components rather than the tumor itself, and these drugs target fibroblast-specific proteins and secreted extracellular matrix residents [2]. The personalized therapy concept could also be extended to the tumor microenvironment, thus providing a more comprehensive way to treat cancer [18]. CAFs produce a tumor-supportive ECM that promotes the growth and dissemination of pre-neoplastic epithelial cells, thereby increasing malignant transformation [19]. CAFs surround tumor cells and can enhance tumor growth through the secretion of growth factors or matrix-degrading enzymes such as MMPs [20,21]. Therefore, a better understanding of the interplay between CAF and tumor cells in relation to the tumor microenvironment will be important in developing strategies for the inhibition of cancer progression. Attempts have been made to elucidate the underlying mechanism of the development of HNSCC based on the characteristics of the cancer cells themselves. However, a clear cancer-associated gene or molecular machinery of the pathogenesis has not yet been investigated. Furthermore, few studies have examined the molecular mechanisms by which CAF in HNSCC regulates cancer progression.

In the present study, NAB2 immunoreactivity was detected in interstitial CAFs of primary and metastatic lymph nodes, whereas tumor cells and NTFs were negative for NAB2 expression. Indeed, the expression of CAF markers and MMPs was higher in CAFs than in paired NTFs isolated from HNSCC patient tissue. Furthermore, NAB2 was more highly expressed in CAFs than in NTFs and its overexpression in the latter resulted in the upregulation of CAF markers and MMPs, whereas NAB2 knockdown had the opposite effect. These results suggest that NAB2 expression characterizes CAFs in HNSCC. It is generally known that MMP expression in CAFs is important for cancer progression, although further studies are needed to clarify the molecular basis for NAB2-dependent MMP upregulation in CAFs.

To investigate the effect of NAB2 derived from CAFs on HNSCC cell invasion, we used a co-culture system to mimic the tumor microenvironment, including 2-D cancer cells and patients;’ CAF in vitro. Indirect co-culture model employing inserted porous membrane to keep the co-cultivated cell populations separated has provided more reproducible results in vitro [22]. A 3-D cell culture system can recapitulate the in vivo cancer microenvironment in studies of cancer cell invasion [23]. Furthermore, previous study showed that the CM in which fibroblasts are grown can enhance the tumorigenicity of breast cancer grafts [7]. In the present study, CM from NAB2-overexpressing CAFs or from NAB2-deficient CAFs enhanced or inhibited FaDu spheroids invasion, respectively. These data suggest that FaDu-CAF acts via NAB2 to increase the malignant potential of FaDu.

Recently, mouse tumor models have been established by transplanting spheroids instead of dissociated cancer cells, since several studies have demonstrated that these more closely recapitulate the disease than two-dimensional cells [24]. We used a conventional xenograft mouse model to investigate the in vivo role of NAB2 expressed by CAFs and found that NAB2 promoted the growth of FaDu spheroid-derived tumors and increased their expression of MMPs and factors associated with cell proliferation or invasion (TGF-β1, *N*-cadherin). NAB2 has primarily been studied as a transcriptional repressor of EGR1. However, in this study, NAB2 function was independent of EGR1. The molecular mechanism by which NAB2 expressed in CAFs promotes HNSCC progression remains to be elucidated in future studies.

Previous data demonstrated that ectopic expression of NAB2 in normal fibroblasts of Scleroderma abrogated TGF-β-induced stimulation of collagen synthesis [25]. Furthermore, mice with targeted deletion of NAB2 displayed increased collagen accumulation in the dermis, suggesting that NAB2 as a novel negative regulator of TGF-β signaling responsible for setting the intensity of fibrotic response [25]. Interestingly, Fang et al. demonstrated that collagen can be a double-edged sword in tumor progression, both inhibiting and promoting tumor progression at different stages of cancer development [26]. However, there are no previous data explaining the expression and function of NAB2 in cancer metastasis, and the in vivo functions of NAB2 are incompletely understood. Our data suggests that the molecular mechanisms by which NAB2 affects ECM in a cancer microenvironment are presumed to be different from the action of NAB2 known in other tissues so far. Further study that examines how NAB2 increases MMP2 in a cancer microenvironment, the expression of CAF markers in fibroblast cells is particularly needed. It is necessary to investigate whether NAB2 regulates cancer progression through collagen in cancer microenvironment. Considering the characteristics of FaDu which is derived from hypopharynx, primary CAF cells were cultured as possible as from the posterior part of the oral cavity nearby oropharynx. However, further studies will also need to determine whether the effect of CAF cultured from oropharyngeal cancer on FaDU is identical to this study.

## 4. Materials and Methods

### 4.1. Immunohistochemical Analysis of Clinical Specimens

Paraffin-embedded tissue blocks from each of the 10 non-metastatic HNSCC tissues, metastatic primary, and paired metastatic lymph node tissues were obtained from patients with HNSCC who underwent tumor resection for oral cancer treatment from 2010–2015 at the Kyungpook National University Hospital. Informed consent was obtained from all patients before the collection of specimens. The study was approved by the Ethics Committee of the Kyungpook National University Hospital (KNUH201501030, the date of approval: 30 January 2015). The patients’ information is as follows: 10 metastatic and 10 non-metastatic patients with average ages of 57.7 ± 12.1 and 61.2 ± 7.8 years, respectively; each group comprised eight males and two females. After dewaxing, sections (5 μm) were blocked for 5 min, then incubated for 2 h at room temperature with NAB2 antibody (1:500). IHC staining was performed using the UltraTek horseradish peroxidase (HRP) Anti-Polyvalent kit (ScyTek Laboratories, Logan, UT, USA); the chromogen used was 3.3-diaminobenzidine (Dako, Carpinteria, CA, USA). Nuclei were counterstained with hematoxylin. Two experienced researchers who were blinded to the origin of the sections evaluated the immunoreactivity independently. NAB2-positive cells were counted in 100 cells of each sample: 10 representative fields at 200× magnification were selected and the cells were counted. Thus, by selecting 10 random areas that were not contiguous with each other, possible errors in recounting the same cell were minimized.

### 4.2. Chemicals and Reagents

Dulbecco’s Modified Eagle’s Medium (DMEM), fetal bovine serum (FBS), and penicillin/streptomycin were obtained from Invitrogen (Carlsbad, CA, USA). Qiazol was from Qiagen (Valencia, CA, USA), and 2× SYBR Green PCR Master Mix was from Takara Bio (Otsu, Japan). Rabbit anti-NAB2 (Santa Cruz Biotechnology Cat# sc-22815, RRID:AB_2298032) and HRP-conjugated anti-β-actin (Santa Cruz Biotechnology Cat# sc-47778 HRP, RRID:AB_2714189) antibodies were from Santa Cruz Biotechnology (Santa Cruz, CA, USA). Rabbit anti-MMP9 (Abcam Cat# 2551-1, RRID:AB_1267245) and mouse anti-MMP2 (Abcam Cat# ab7032, RRID:AB_2145819) antibodies were purchased from Abcam (Cambridge, MA, USA). Rabbit anti-α-SMA and -MMP14 (Proteintech Group Cat# 14552-1-AP, RRID:AB_2250751) antibodies were from Proteintech (Rosemont, IL, USA). Rabbit anti-fibroblast activation protein (FAP) antibody was from GeneTex (GeneTex, Inc., Irvine, CA, USA).

### 4.3. Primary Fibroblast Cultures and Preparation of Conditioned Medium (CM)

HNSCC and surrounding non-tumor oral tissues were obtained by surgical resection from three patients (P1, P2, and P3; Table A1) at Kyungbook National University Hospital. None of the patients had received radiotherapy or chemotherapy prior to surgery. The specimens were used after obtaining written, informed consent from the patients and with the approval of the Institutional Research Ethics Committee of Kyungbook National University Hospital (KNUH201704011, the date of approval: 11 April 2017). The FaDu human HNSCC cell line was obtained from American Type Culture Collection (ATCC Cat# HTB-43, RRID:CVCL_1218) and cultured in DMEM containing 10% FBS and 1% penicillin/streptomycin solution at 37 °C in a 5% CO_2_ humidified atmosphere.

The stroma adjacent to the cancer or normal tissue was separated carefully by a pathologist and were cut into the smallest possible pieces in sterile DMEM and were seeded in 10-cm tissue culture dishes supplemented with 10% FBS. They were incubated for 24 h to allow attachment to the culture plate, and the unattached cells were removed. After 3 passages, primary CAFs and paired non-tumor fibroblasts (NTFs) were grown for 48 h until they reached 80% confluence. The culture medium was collected and passed through a 0.45 μM pore filter; the filtrate was used as the CM. Cells from passages 3–6 were used for all experiments.

### 4.4. Real-Time Quantitative Polymerase Chain Reaction (qPCR)

Total RNA extraction, cDNA synthesis, and gene expression normalization were carried out according to standard protocols [27]. The following forward and reverse primers were used for real-time quantitative polymerase chain reaction (qPCR): NAB2, 5′-CACATCCCTGCTAAAGCTGAA-3′ and 5′-GTCGAAACGGCCATAGATGAT-3′; MMP2, 5′-AGCTGCAACCTGTTTGTGCTG-3′ and 5′-CGCATGGTCTCGATGGTATTCT-3′; MMP9, 5′-ACGACGTCTTCCAGTACCGAGA-3′ and 5′-TAGGTCACGTAGCCCACTTGGT-3′; MMP14, 5′-CGAGGTGCCCTATGCCTAC-3′ and 5′-CTCGGCAGAGTCAAAGTGG-3′; α-SMA, 5′-GTGTTGCCCCTGAAGAGCAT-3′ and 5′-GCTGGGACATTGAAAGTCTCA-3′, FAP, 5′-CAAAGGCTGGAGCTAAGAATCC-3′ and 5′-ACTGCAAACATACTCGTTCATCA-3′; transforming growth factor (TGF)-β, 5′-ACCTGAACCCGTGTTGCTCT-3′ and 5′-GCTGAGGTATCGCCAGGAAT-3′; N-cadherin, 5′-GGTGGAGGAGAAGAAGACCAG-3′ and 5′-GGCATCAGGCTCCACAGT-3′; vimentin, 5′-GCAAAGCAGGAGTCCACTGAGT-3′ and 5′-ATTTCACGCTCTGGCGTTC-3′; and glyceraldehyde 3-phosphate dehydrogenase (GAPDH), 5′-AGATCATCAGCAATGCCTCCTG-3′ and 5′-CTGGGCAGGGCTTATTCCTTTTCT-3′. Gene expression levels were normalized to that of the housekeeping gene GAPDH. qPCR was carried out on an ABI 7500 real-time PCR system (Applied Biosystems, Foster City, CA, USA).

### 4.5. Western Blot Analysis

Total protein was extracted and the concentration was determined as previously described [27]. Equal amounts of protein (30–40 µg) were separated by 8–15% sodium dodecyl sulfate–polyacrylamide gel electrophoresis (SD–PAGE) and then transferred to a nitrocellulose membrane, blocked in 5% skim milk for 2 h, and incubated overnight at 4 °C with primary antibodies; β-actin served as the loading control. HRP-conjugated secondary antibodies were applied at 1:5000 dilutions for 1 h at room temperature, and the blot was washed three times in Tris-buffered saline containing 0.1% Tween 20. Protein bands were detected by enhanced chemiluminescence and their relative intensities were analyzed using ImageJ software (National Institutes of Health, Bethesda, MD, USA).

### 4.6. Transfection of Small Interfering (si)RNA or Overexpression Vector

Fibroblasts were transiently transfected with a siRNA mixture targeting the NAB2 transcript (siNAB2; Santa Cruz Biotechnology). The cells (1 × 10^5^) were seeded on a 60 mm plate, and the following day the medium was replaced with serum-free medium immediately before transfection with control siRNA or siNAB2 at a final concentration of 10 nM using Lipofectamine 2000 (Thermo Fisher Scientific, Waltham, MA, USA). After 8 h, the medium was replaced with fresh serum-containing medium. For NAB2 overexpression, cells were transfected with pcMV6-AC-tGFP NAB2 (Origen Technologies, Rockville, MD, USA) or with the empty vector as a control.

### 4.7. Matrigel Invasion Assay

The invasion of FaDu cells was evaluated in a co-culture system using Matrigel-coated 8.0 μm filter chambers (BD Biosciences, San Jose, CA, USA). CAFs were transfected with NAB2 overexpression or control vector and seeded in a 24-well plate (Figure 4A). FaDu cells were resuspended in medium and 300 µL aliquots were added to each Matrigel-coated transwell. After culturing for 48 h, the cells were stained with 0.2% crystal violet in 10% ethanol; those on the upper side of the transwell insert were removed with a cotton swab, while cells that had migrated to the lower surface of the filter were counted. The invasion index was calculated as the fold change in the number of invaded cells in the experimental group compared to that in the control group.

We also evaluated the effect of CM from cultures of CAFs transfected with siNAB2, NAB2, or the respective control vectors on three-dimensional (3-D) FaDu spheroid invasion [24,28]. Briefly, FaDu cells were seeded on a 96-well U-bottom ultra-low attachment plate (5000 cells/well) (Corning Inc., Corning, NY, USA) and cultured for 3 days until spheroids had formed (>500 μm in diameter). After replacing the medium with 100 μL CM, 50 μL Matrigel were added to each well to provide a semi-solid matrix; when it had solidified, 100 μL CM were added more to prevent drying. Invasion was monitored over a period of 14 days by phase-contrast microscopy (5× magnification) and quantified by measuring the average length and number of tube-like structures extending from the surface of each spheroid. MMP mRNA expression in FaDu spheroids was analyzed by qPCR on day 14.

### 4.8. Mouse Xenograft Model

We evaluated the effect of NAB2-overexpressing CAFs on FaDu spheroid-derived tumor growth in xenograft mice (6-week-old female BALB/c; Hyochang Science, Daegu, Korea). All experimental protocols followed the ARRIVE guidelines (Animal Research: Reporting of In Vivo Experiments) and were approved by the Animal Ethics Committee of Kyungpook National University (KNU 2017-94-2, the date (6 September 2017) of approval). FaDu spheroids were prepared in 96-well U-bottom ultra-low attachment plates (<400 μm in diameter). CAFs derived from P3 were transfected with control or NAB2 overexpression vector for 2 days. A total of 50 FaDu spheroids and 5 × 10^5^ CAFs were co-injected into the 5 mice using a 22-gauge needle. To reduce error caused by variability among mice, CAFs transfected with the control or NAB2 overexpression vector were co-injected with FaDu spheroids into the oral mucosa of the right and left cheeks, respectively. After 6 weeks, mice were sacrificed and tumor volume was measured using clipper. Tumor histology was examined by H&E staining. MMP expression from tumor tissues was analyzed by qPCR.

### 4.9. Statistical Analysis

Differences between groups were evaluated with the parametric two-tailed non-paired *t* test. Analyses were performed using Origin v.8.0 software (OriginLab, Northampton, MA, USA), and *p* values ≤ 0.05 were considered statistically significant.

## 5. Conclusions

The results of the present study demonstrate that oral CAFs express higher levels of NAB2 than NTFs and that this promotes HNSCC progression via upregulation of MMPs. These findings provide insight into the molecular mechanisms underlying the tumor-promoting functions of CAFs, and suggest NAB2 as a potential therapeutic target for preventing the malignant transformation and metastatic progression of HNSCC.

## Figures and Tables

**Figure 1 cancers-11-00388-f001:**
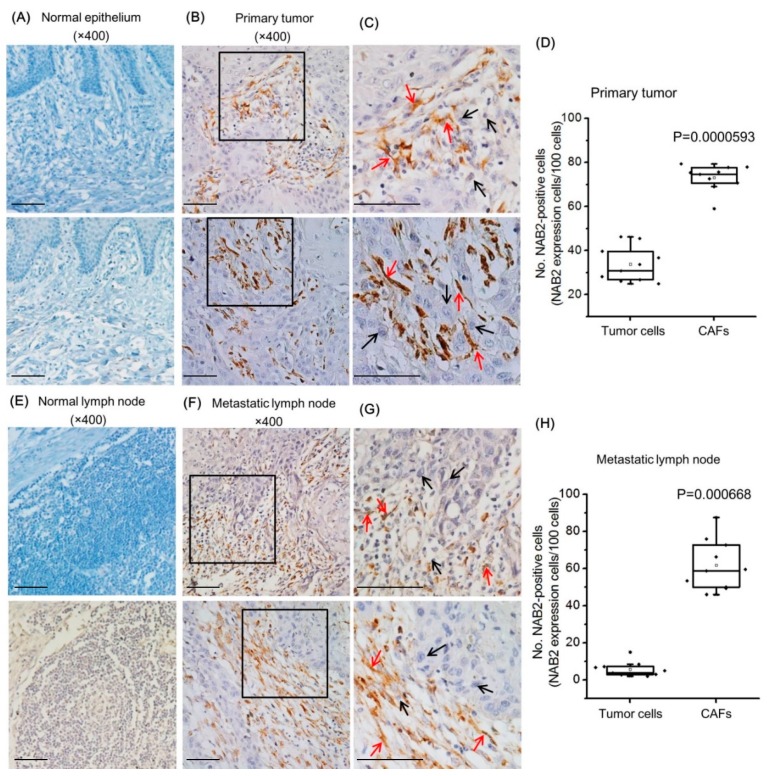
NAB2 expression in head and neck squamous cell carcinoma (HNSCC) patient tissues. NAB2 protein expression was compared in normal epithelial cells (**A**) and primary tumors (**B**,**C**), and normal lymph nodes (Scale bar in C: 100 μM) (**E**) and paired metastatic lymph nodes (**F**,**G**) by immunohistochemistry. (**C**) and (**G**) are two-fold magnifications of the black rectangle images in (**B**) and (**F**), respectively. (Scale bar in G: 100 μM) Nuclei were counterstained with hematoxylin. Red arrows indicate some representative NAB2-expressing interstitial cancer-associated fibroblasts (CAFs) of primary and metastatic lymph node tissues. Black arrows indicate NAB2-negative cancer cells. NAB2-positive tumor cells and CAFs were counted in primary tumor tissues (**D**) and metastatic lymph node tissues (**H**). The boxes show the upper 75% and lower 25% quartiles of the measurements with respect to the median value (horizontal line in each box). Each dot represents an individual patient’s data, while a small open rectangle represents the mean value. Each bar represents the standard deviation.

**Figure 2 cancers-11-00388-f002:**
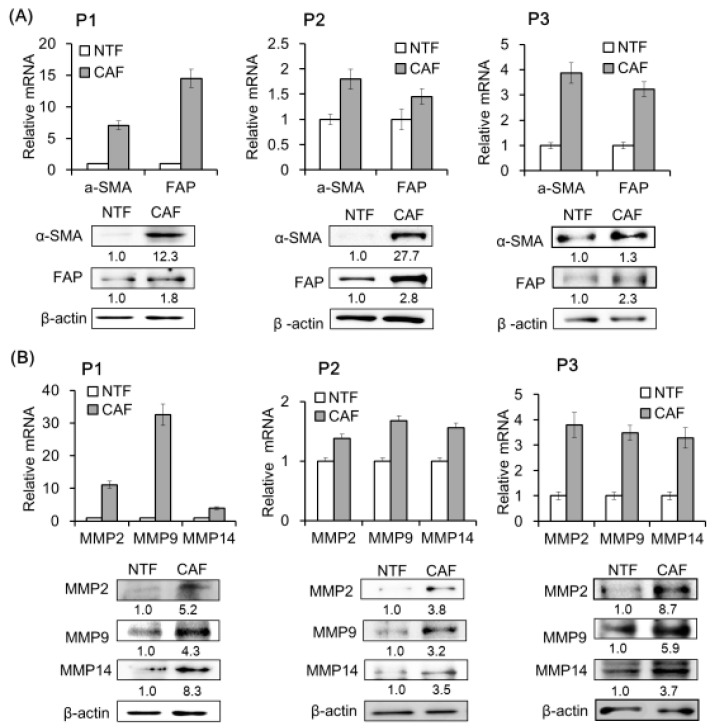
CAF marker and matrix metalloproteinase (MMP) expression in primary fibroblasts from HNSCC patient tissue. (**A**) CAFs and paired non-tumor fibroblasts (NTFs) were isolated from three HNSCC patient tissue (P1, P2, P3) specimens by short-term primary culture. α-SMA and FAP mRNA and protein levels were evaluated by real-time quantitative polymerase chain reaction (qPCR) and Western blotting, respectively. (**B**) MMP mRNA and protein expression was analyzed under the same conditions. Results represent the mean ± standard deviation of two or three experiments.

**Figure 3 cancers-11-00388-f003:**
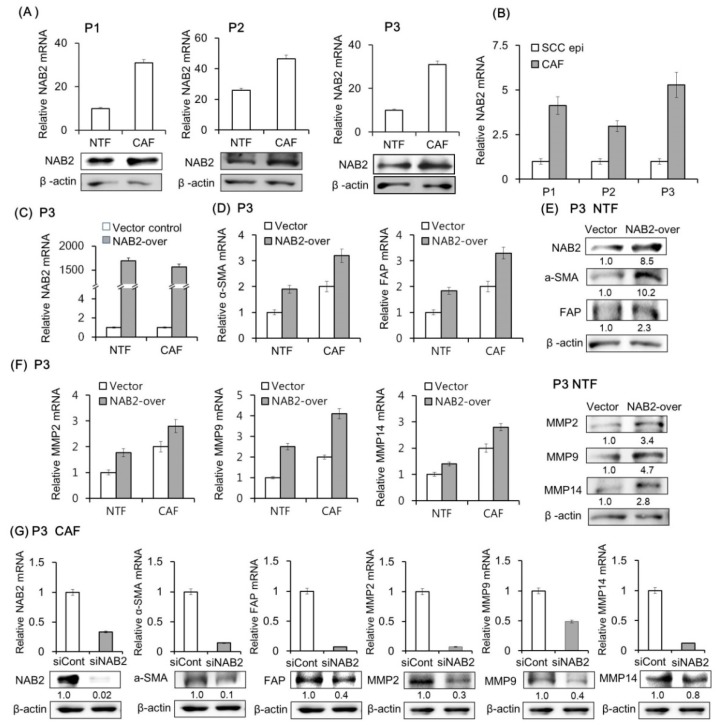
Effect of NAB2 on CAF marker and MMP expression in patient fibroblasts. (**A**) NAB2 mRNA and protein expression in CAFs and paired NTFs from tissue specimens of three HNSCC patients was analyzed by qPCR and Western blotting, respectively. (**B**) NAB2 mRNA level is compared in cultured CAFs and paired HNSCC tissues. (**C**) CAFs and NTFs form P3 were transfected with NAB2 overexpression (NAB2-over) or control vector for 48 h. (**D**,**E**) mRNA and protein levels of CAF markers were analyzed in NTFs transfected with NAB2-over or control vector. (**F**) mRNA and protein levels of MMPs were analyzed under the same conditions. (**G**) CAFs were transfected with control siRNA or siNAB2 mixture for 48 h, and mRNA and protein levels of NAB2, CAF markers, and MMPs were analyzed. Results represent the mean ± standard deviation of two or three experiments.

**Figure 4 cancers-11-00388-f004:**
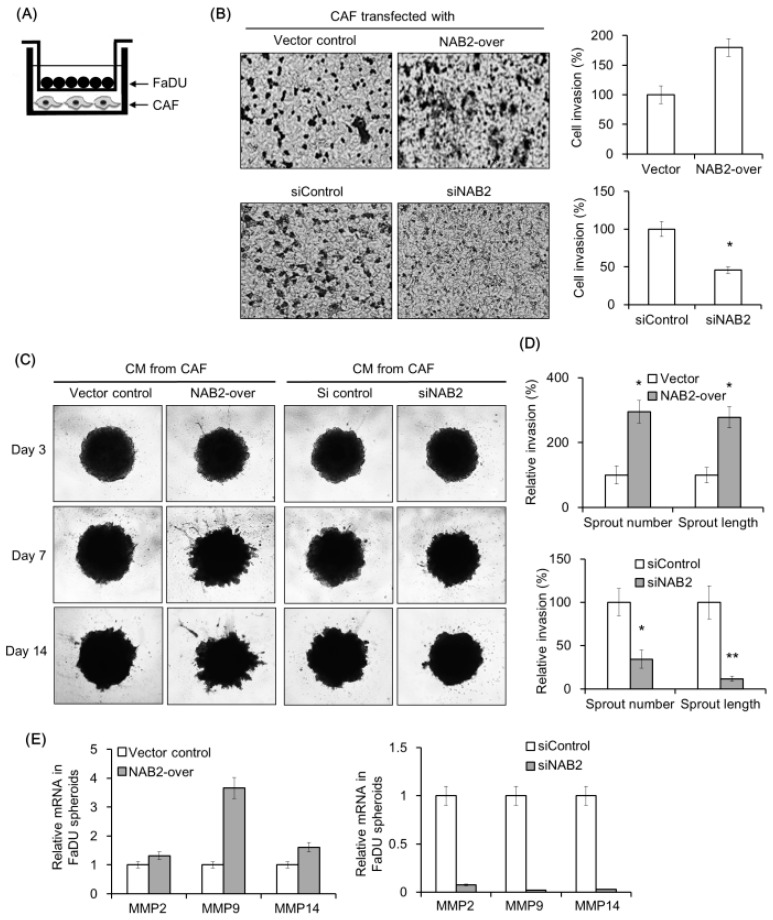
Effect of NAB2 expressed by CAFs on FaDu cell or spheroid invasion. (**A**) CAFs were transfected with NAB2 overexpression or control vector and seeded in a bottom well of the plate. FaDu cells were added to the Matrigel-coated transwell. (**B**) After culturing for 48 h, cells in the transwell were stained with crystal violet and those that had migrated to the lower surface of the transwell insert were counted (5× magnification). Cell invasion index was calculated as the difference in the number invaded cells between the NAB2 overexpression and control groups. The effects of siNAB2 or control siRNA transfection were compared under the same conditions. To evaluate the effect of conditioned medium (CM) from CAF cultures on FaDu spheroid invasion, short-term primary culture of CAFs and paired NTFs were transfected with NAB2 overexpression or control vector, or with siNAB2 or control siRNA. After 48 h, the culture supernatant was collected from each plate and passed through a 0.45 μM filter and used as CM. (**C**) FaDu spheroid (>500 μm in diameter) was formed by culturing in a 96-well U-bottom ultra-low attachment plate (5000 cells/well) for 3 days. Matrigel (50 μL) was added to each well containing 100 μL CM, and spheroid invasion was monitored for 14 days by phase-contrast microscopy (5× magnification). (**D**) Cell invasion was quantified by measuring the average length or number of tube-like structures extending from the surface of each spheroid. (**E**) MMP mRNA expression in FaDu spheroids was analyzed by qPCR on day 14. Results represent the mean ± standard deviation of two or three experiments. * *p* < 0.05, ** *p* < 0.01.

**Figure 5 cancers-11-00388-f005:**
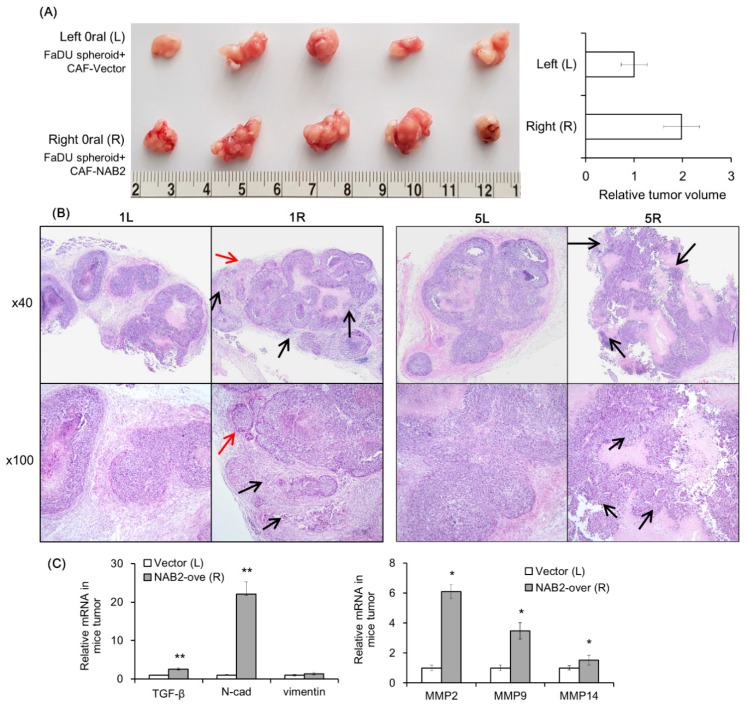
Effect of NAB2-overexpressing CAFs on the growth of FaDu spheroid-derived tumors in nude mice. FaDu spheroids (>400 μm in diameter) were prepared in 96-well U-bottom ultra-low attachment plates. CAFs from P3 transfected with NAB2 overexpression or control vector; 50 FaDu spheroids and 5 × 10^5^ CAFs transfected with NAB2 overexpression or control vector were co-injected into the oral mucosa of the left and right cheeks, respectively, of BALB/c mice. (**A**) Mice were sacrificed after 6 weeks and tumor size was measured. (**B**) Hematoxylin and eosin (H&E) staining was performed using tumor tissues from nude mice. The red and black arrows indicate some representative lobulated margins and increased invasion into the surrounding connective tissue in tumor, respectively. (**C**) mRNA expression of MMPs and invasion-related genes in tumor tissues was investigated at the same time. * *p* < 0.05, ** *p* < 0.01.

**Figure 6 cancers-11-00388-f006:**
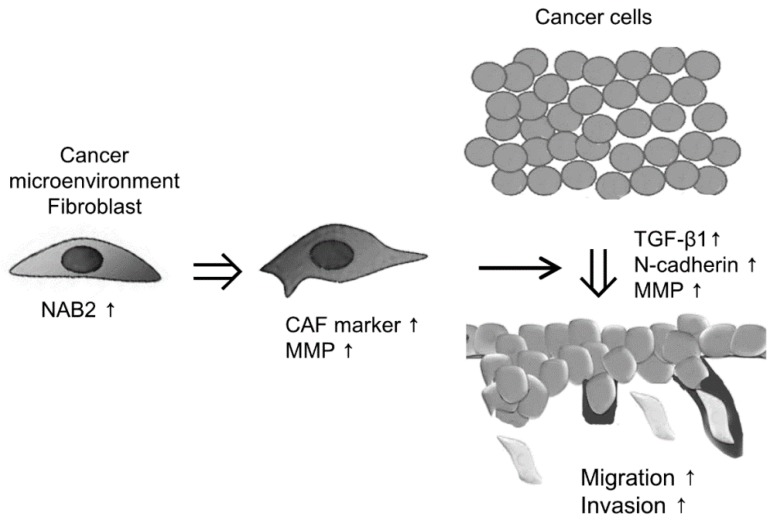
Putative schematic of HNSCC progression by NAB2 derived from CAF. Results represent the mean ± standard deviation of two experiments.

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
