# Peer review of "NAB 2-Expressing Cancer-Associated Fibroblast Promotes HNSCC Progression"

_cancers, 2019, doi:10.3390/cancers11030388_

Round 1
Reviewer 1 Report
The manuscript is very interesting. But, authors must improved the quality of the manuscript.
FaDu is hypopharyngeal cancer cell line. Namely, FaDu is not oral squamous cell carcinoma. Therefore, authors must revise the title etc.
Therefore, the experiment using nude mice is not orthotopic.
In line22, authors stated that CM of NAB2-overexpressing CAFs increased the invasion of FaDu as compared to CM of NTF. However, I can see that NAB2-overexpressing CAFs increased the invasion of FaDu as compared to CM of control CAF in Figure4(B).
In line 212, authors must revise the following sentence "cancer development (reviewed in [25]."
Author Response
The manuscript is very interesting. But, authors must improve the quality of the manuscript.
FaDu is hypopharyngeal cancer cell line. Namely, FaDu is not oral squamous cell carcinoma. Therefore, authors must revise the title etc.
Therefore, the experiment using nude mice is not orthotopic.
Response: We regret our mistakes. All OSCCs have been modified to head and neck squamous cell carcinoma (HNSCC).In addition, the word ‘orthotopic’ was deleted. Thank you for the kind comment.
In line22, authors stated that CM of NAB2-overexpressing CAFs increased the invasion of FaDu as compared to CM of NTF. However, I can see that NAB2-overexpressing CAFs increased the invasion of FaDu as compared to CM of control CAF in Figure 4(B).
Response: Thank you for the comment. Figure 4B shows the Matrigel invasion of FaDu cells under co-culture with CAFs transfected with control vector or NAB2-overexpression vector, or with control siRNA or siNAB2. We modified this sentence as follows.
The invasion of FaDu cells was markedly increased under co-culture with NAB2-overexpressing CAFs as compared to control vector-transfected CAFs (Fig. 4B).
In line 212, authors must revise the following sentence "cancer development (reviewed in [25]."
Response: We revised this sentence and inserted the bibliography into the correct form.
Reviewer 2 Report
The authors demonstrated that CAFs with higher expression of NAB2 promoted OSCC progression through up-regulation of MMPs in vitro and in vivo. They suggested NAB2 as a potential therapeutic target for preventing the malignant transformation and metastatic progression of OSCC. Although the mechanisms of MMPs induction by NAB2 remain unclear, their findings are novel and interesting for oncologists. Some points should be reconsidered.
Additional NAB2 IHC staining should be performed (maybe over 50 samples). And It should be statistically proved that CAFs express NAB2 strongly compared to normal epithelium or cancer cells using OSCC patient tissues.
Figure3C, there are not columns of Vector.
Figure 2, 3, 4, most columns of the graphs do not have asterisks (only Figure 4B, 4D and 5C show asterisks). The authors should show that the difference is statistically significant. Or does it mean duplicate or triplicate of qPCR using one each RNA sample from P1, P2, or P3? The details should be described.
In supplementary table1, all patients had lymph node metastases. Then their TNM stage must not be N0.
Author Response
The authors demonstrated that CAFs with higher expression of NAB2 promoted OSCC progression through up-regulation of MMPs in vitro and in vivo. They suggested NAB2 as a potential therapeutic target for preventing the malignant transformation and metastatic progression of OSCC. Although the mechanisms of MMPs induction by NAB2 remain unclear, their findings are novel and interesting for oncologists. Some points should be reconsidered.
Additional NAB2 IHC staining should be performed (maybe over 50 samples). And it should be statistically proved that CAFs express NAB2 strongly compared to normal epithelium or cancer cells using OSCC patient tissues.
Response: Thank you for the critical comment. First, NAB2 staining was quantified for 30 previously stained slides. In fact, the pattern of CAF-specific NAB2 immunoreactivity was very clear. Also, since the deadline for revising this manuscript (10 days) was tight, it was hard to proceed with more slides. If you do not mind, please take a look at our revision. If you would still like us to analyze more slides, we are willing to extend the revision due date and back it up with more slides.I seek your understanding for this situation. The cell counting graphs are shown in Fig. 1D and Fig. 1H.
Figure3C, there are not columns of Vector.
Response: As you pointed out, the overexpression efficiency of NAB2 was too high compared to the vector control, resulting in a relatively low control value. We modified Fig. 3C to take this into account.
Figure 2, 3, 4, most columns of the graphs do not have asterisks (only Figure 4B, 4D and 5C show asterisks). The authors should show that the difference is statistically significant. Or does it mean duplicate or triplicate of qPCR using one each RNA sample from P1, P2, or P3? The details should be described.
Response: The PCR data are shown by triplicated analysis of each RNA sample. We performed three biologically independent PCR analyses with each RNA sample. Although there was a slight difference in the fold induction values, the expression patterns were all the same and the results were summarized like this. Therefore, no P-value was added.
In supplementary table1, all patients had lymph node metastases. Then their TNM stage must not be N0.
Response: Thank you for the kind suggestion. We deleted the column for lymph node metastasis.
Round 2
Reviewer 1 Report
Unfortunately, this manuscript needs major revision. FaDu is hypopharyngeal cancer cell lines. But, authors used oral CAF. Hypopharynx is far from oral cavity. When authors used oral CAF, they should use oral SCC cell lines. When authors used FaDu, they should use hypopharyngeal CAF.
HNSCC involves some subsites, ie, oral cavity, nasopharynx, oropharynx, hypopharynx, larynx, etc. They have different characteristic.
Author Response
Unfortunately, this manuscript needs major revision. FaDu is hypopharyngeal cancer cell lines. But, authors used oral CAF. Hypopharynx is far from oral cavity. When authors used oral CAF, they should use oral SCC cell lines. When authors used FaDu, they should use hypopharyngeal CAF.
HNSCC involves some subsites, ie, oral cavity, nasopharynx, oropharynx, hypopharynx, larynx, etc. They have different characteristic.
Response: First of all, thank you for your interest in this study. I express deep respect for your rigorous research mindset, and I fully understand your concern on this study. Please let me explain the reason we chose FaDu cell line for this study. The trend of in vitro cancer research has been shifting toward 3-dimensional spheroid which is close to in vivo as it mimics the cancer microenvironment more than dissociated cells. Matrigel invasion of cancer cells, which is one of the main topics in this study, was also an important selection criterion. Therefore, when conducting preliminary experiments on various OSCC cell lines including UMSCC1, YD-10B, and other CAL series, the results were not satisfactory. FaDu, as noted, is a cell line established from a pharynx cancer patient's tissue of poorly differentiated grade, and the formation of the spheroid is very successful and reproducible. In addition, in vitro Matrigel invasion appeared well in both 2-D and 3-D cultures, suggesting that it is well suited to this experiment. We also tried to observe the effect of NAB2 expressed in CAF by modulating NAB2 gene on in vivo carcinogenesis. For this experiment, it is not easy to establish a stable cell line with primary cultured CAF cells. In this regard, FaDu was considered to be the optimal cell line for this study due to its rapid tumorigenicity.
Unfortunately, we could not use pharyngeal cancer tissues since we first started this study with IRB approval for oral cavity tissue retrieval to isolate CAF. Thus, we searched existing papers to determine whether there is any disagreement in the use of FaDu for OSCC studies. In many of the previous OSCC studies, FaDu was interchangeably used in vitro and in vivo study with cell lines derived from the oral cavity. Thus, in this study, we decided to perform CAF-related studies derived from the oral cavity using FaDu cell. Considering the characteristics of FaDu, primary CAF cells were cultured as possible as from the posterior part of the oral cavity nearby oropharynx. We hope this will help you understand our best interests, and have your concerns addressed in the Discussion section so that other researchers can refer this point to their study. This is our best, but thank you again for your important comments. It will be very helpful in our study in the future.
Round 3
Reviewer 1 Report
From line235 to 238, FaDu is derived from hypopharynx., primary CAF cells were cultured as possible as from the posterior part of the oral cavity nearby oropharnx. However, hypopharynx is different from oropharynx. Furthermore, recently, oropharyngeal cancer divided into two categories, HPV positive and HPV negative. Therefore, it is not useful that primary CAF cells were cultured as possible as from the posterior part of the oral cavity nearby oropharnx, when authors used hypopharyngeal cancer cell lines.
Author Response
Please find the responses in the attachment. Thanks.
